# Determinants of tea planters' purchasing behavior of planting insurance: SEM analysis

**Zhenhua Xu, Nuttawut Rojniruttikul**[ID]*

KMITL Business School, King Mongkut's Institute of Technology Ladkrabang

* nuttawut.ro@kmitl.ac.th

## Abstract

This study explores the influence of government support and risk cognition on tea planters' purchasing behavior regarding planting insurance, with a particular emphasis on the mediating roles of perceived value and environmental concern. Data were collected from 550 tea planters in Guizhou Province, China, using a structured questionnaire and convenience sampling method. Confirmatory factor analysis (CFA) and structural equation modeling (SEM) were conducted using AMOS 28.0 to analyze the data. The results indicate that both government support and risk cognition significantly and positively impact the perceived value of planting insurance. To promote the uptake of planting insurance among tea farmers, government agencies should strengthen policy advocacy and provide business guidance. Such efforts would help tea farmers recognize the value and psychological significance of planting insurance, thereby better safeguarding their agricultural interests. This study enhances the understanding of how government support and risk cognition can facilitate the adoption of planting insurance among tea farmers.

## 1. Introduction

### 1.1 Background

China, as the leading global producer of tea, accounted for 3.25 million tons of production in 2023, representing 49.21% of the world's total output. Other major tea-producing nations include India, Kenya, and Sri Lanka [1]. Since 2014, China's tea cultivation area has expanded steadily, reaching approximately 52.50 million acres by 2024, marking a significant increase from 41.636 million acres in 2013 [2].

Despite its economic advantages, tea cultivation is highly vulnerable to natural disasters, resulting in substantial financial losses [3]. In response to this vulnerability, tea planters are increasingly opting for planting insurance, specifically designed for cash crops. This type of insurance is notable for several reasons: it protects against a range of natural hazards that may affect tea production, offers significant financial compensation in the event of crop damage, and benefits from substantial government subsidies and policy incentives. As a result, tea planters widely adopt this insurance model [4].

**Data availability statement:** The datasets generated and analyzed during the current study were collected with informed consent from all survey participants. Due to ethical restrictions imposed by the Ethics Committee of Tongren Polytechnic College, the minimal dataset cannot be publicly shared. However, the data are available for academic use upon reasonable request. Interested researchers may contact Dr. Zhao Wei, Research Affairs Coordinator at the Scientific Research Department, via email at trzykyzw@163.com for data access queries. Methodology details, including the validated questionnaire and sampling framework, are comprehensively described in the paper.

**Funding:** The author(s) received no specific funding for this work.

**Competing interests:** The authors have declared that no competing interests exist.

Agricultural insurance has become a key strategy for mitigating risks in farming, especially in mountainous regions. Its success has laid the foundation for extending similar models to other valuable crops, as well as livestock and aquaculture [5]. Serving as a market-driven tool, agricultural insurance enables farmers to reduce agricultural uncertainties. However, to maximize its potential—particularly in increasing farmers' incomes—enhanced participation and sustained commitment to insurance schemes are essential.

Several barriers hinder the effectiveness of agricultural insurance. High premium costs can be a significant burden for financially constrained farmers, and coverage limits may not sufficiently protect against potential losses. Furthermore, a lack of understanding of how insurance operates can prevent farmers from recognizing its benefits. Inefficient subsidy distribution often fails to reach the most vulnerable farmers. Additionally, over-reliance on premium subsidies without addressing these factors can lead to unintended negative consequences.

Farmers' perception of risk plays a crucial role in determining their willingness to engage with agricultural insurance. The perceived value of insurance is pivotal in shaping their purchasing decisions [6]. However, challenges remain in this area. Farmers often struggle to understand complex government support policies, leading to reduced confidence and participation. Misunderstandings about insurance products, including ambiguous terms and benefits, contribute to hesitance. Moreover, many farmers are unfamiliar with the claims process and broader risk management strategies, further complicating their engagement with insurance. Misinterpretations—such as equating insurance with gambling or viewing it as a government handout—also distort farmers' understanding of its value.

According to the "China Agricultural Insurance Market Demand Survey Report," Only 14.61 percent fully understood the agricultural insurance terms [1].

Bridging these knowledge gaps and enhancing farmers' understanding of risk management are essential to maximizing the impact of government support and fostering a risk-aware agricultural culture.

Although market inefficiencies pose challenges to the success of commercial agricultural insurance, many countries address this issue by providing financial subsidies to stabilize the market and improve farmers' purchasing power [6]. In the context of policy-based agricultural insurance, governments employ various financial aid strategies [7]. However, over-reliance on subsidies can raise concerns about long-term fiscal sustainability and lead to unintended outcomes. Thus, adopting a "smart" subsidy model is crucial.

Government intervention is necessary in agricultural insurance due to potential market failures in private insurance systems. However, merely increasing subsidies is insufficient. A differentiated subsidy approach tailored to the specific needs of various farm sizes and crop types can promote equitable and sustainable growth [8].

While supportive regulations and policies are essential for building a thriving agricultural insurance market, a major challenge lies in the absence of a comprehensive national policy framework. Strong government oversight is vital to ensuring fair practices and protecting farmers [9].

Farmers' participation in agricultural insurance is influenced by a combination of environmental, social, and psychological factors. Their attitudes toward agricultural risks drive their risk management decisions, which can be categorized into "perceived risk"—understood through emotional and intuitive responses—and "analytical risk," based on logical and data-driven evaluations of potential hazards [10].

However, gaps in knowledge and access to information often lead farmers to act in ways that deviate from purely rational behavior. Simon's theory of "bounded rationality" suggests that farmers rely on intuition and personal experience when assessing risks, influenced by both internal psychological factors and external environmental conditions.

The participation patterns of tea farmers in insurance schemes provide a valuable framework for understanding the dynamics of insurance adoption in other high-value cash crops. Consequently, studying tea planting insurance has broader implications for developing policies that enhance insurance uptake across various agricultural sectors.

This research uses tea planting insurance as a case study to explore how government support and risk perception jointly influence farmers' decisions to purchase insurance. Additionally, it examines the role of perceived value, environmental awareness, and the pathways through which these factors exert influence. The findings offer both theoretical and practical insights that can inform the design and optimization of agricultural insurance policies, ultimately enhancing this critical risk management tool for diverse crops.

### 1.2 Research objectives

The objectives of this study aim to comprehensively understand the interaction between government support, risk cognition, perceived value, and environmental concern in influencing tea planters' purchasing decisions for planting insurance.

Objective 1: Empirically assess the impact of government support, risk cognition, perceived value, and environmental concern on tea planters' purchasing behavior regarding planting insurance.

Objective 2: Investigate the mediating role of perceived value in the relationship between government support, risk cognition, and tea planters' purchasing behavior.

Objective 3: Examine the mediating effect of environmental concern on the relationship between government support, risk cognition, and tea planters' purchasing behavior.

Objective 4: Develop a behavioral promotion model to enhance tea planters' participation in planting insurance.

## 2. Literature review.

### 2.1 Government support

Government support plays a crucial role in areas such as scientific advancement, environmental sustainability, agricultural restructuring [11], and the promotion of agricultural technologies [12]. Extensive research on governmental involvement in agricultural insurance development highlights its key functions in formulating regulations, implementing strategic initiatives, and overseeing market operations [13]. While direct government management and substantial financial subsidies are important, the social return from premium subsidies is often considered limited. To enhance government engagement, efforts should focus on closing information gaps, improving regulatory oversight of insurance companies, and ensuring greater clarity in insurance contracts [12].

Government involvement in agricultural insurance spans economic, legal, and administrative dimensions. This includes establishing and refining the legal framework, optimizing financial support policies, improving policy coordination, and fostering the development of public service organizations within the insurance sector [14]. The central government is responsible for setting overarching policies, directions, and strategies, while local governments ensure proper market regulation and play a critical role in the high-quality development of agricultural insurance [15].

Government intervention is crucial to ensure precise and targeted compensation for agricultural losses. With its prearranged contractual structure and technical methodology for loss evaluation, insurance provides a systematic mechanism

for managing risk. Although disaster relief and social insurance remain important tools, integrating agricultural insurance with disaster risk reduction and broader risk management strategies can lead to a more organized and proactive response to agricultural challenges, reducing the need for emergency interventions [16].

## 2.2 Risk cognition

Understanding agricultural risk cognition is key to effective risk management, with agricultural insurance serving as a vital tool for transferring risk. Both theoretical and empirical studies have established the link between farmers' risk cognition and their decision-making processes [17,18,19]. Farmers' attitudes toward risk play a crucial role in shaping their risk management behaviors. These attitudes—defined as preferences or responses in uncertain situations—significantly influence their actions [20,21,22].

Farmers' risk attitudes are generally categorized as risk-seeking, risk-neutral, or risk-averse [23,24]. Additional dimensions such as rationality, risk aversion, and risk pursuit have been identified using utility function curves [25]. When adopting agricultural technologies, farmers may be characterized as conservative or risk-tolerant, with many falling between these extremes [26]. Several methods exist to measure risk attitudes, including the abstract model, which incorporates rational and irrational decision functions [27]. The scenario simulation method, which assesses risk attitudes through hypothetical investment scenarios, is simpler and widely used in studies involving farmers[28].

Risk-averse farmers often employ risk avoidance strategies based on their production experience, combining these with post-event financial measures. Their approach to risk management is shaped by both their understanding of risk and their attitudes toward it. For instance, they might diversify operations or adjust production methods to ensure stable income during periods of uncertainty [29]. While informal strategies, such as relying on social networks or loans, are commonly used to manage risks, reliance on formal mechanisms like agricultural insurance is less prevalent. Farmers tend to use a combination of self-help, government aid, and agricultural insurance to mitigate losses [30]. In sectors such as crop production, where risks like drought and disease are prominent, farmers' risk cognition aligns closely with insurance coverage, influencing their insurance purchasing behavior. Research shows a weak positive correlation between risk awareness and insurance uptake [31].

In conclusion, risk cognition is a critical factor in understanding farmers' decisions to purchase agricultural insurance [32], and willingness to insure is a prerequisite for actual insurance participation [33]. Theoretically, heightened risk cognition correlates with increased willingness to engage with insurance schemes [34].

Several research paradigms are essential in studying risk cognition, including the psychometric, socio-cultural, cognitive, and social amplification paradigms. The "psychometric paradigm" suggests that individuals respond to risks differently depending on their environment, with psychological traits, cultural norms, and institutional factors influencing their risk cognition. This model allows for quantitative analysis, as demonstrated by Slovic (2005), who used surveys to assess individual risk cognitions in specific contexts. In China, this paradigm has been applied to study public risk cognition during natural disasters, such as earthquakes [35].

The "socio-cultural paradigm" examines how social norms, values, and cultural differences shape public risk perception across various groups and regions [36]. It explores the influence of societal values on individuals' perception of risks [37]. The "cognitive paradigm" focuses on how individuals, limited by rationality, exhibit cognitive biases when confronted with uncertain risks. These biases are shaped by internal factors like personality and experience, as well as external factors such as the nature and magnitude of the risk [38].

Finally, the "social amplification of risk paradigm" explores how psychological, social, cultural, and institutional factors interact with risk events, amplifying or diminishing public reactions. Through various channels, including media, risks can be "amplified" to create ripple effects within society [37,39]. This framework is particularly useful for understanding how societal and environmental factors influence public risk cognitions, such as the amplification of environmental risks by external systems [40].

## 2.3 Perceived value

In the early stages of research on customer perceived value, product quality and price were identified as the primary contributors to value. Product quality represented the "gain" aspect, while price was associated with the "cost" or "sacrifice" factor [41]. Zeithaml's model of customer value perception divides value into two main components: "gains" and "costs." The "gains" include elements such as internal and external attributes, perceived quality, and intangible factors like corporate reputation and brand image. On the other hand, the "costs" comprise both monetary and non-monetary factors, such as time, energy, and effort.

To maintain a competitive advantage, businesses must go beyond merely offering high-quality products at fair prices. Delivering exceptional service quality, which is difficult for competitors to replicate, becomes a key factor in achieving sustainable market success [42]. More recent studies continue to emphasize that product quality, service quality, and pricing remain central drivers when evaluating customer perceived value through the lens of gains and losses [43].

However, the combination of product quality, service quality, and pricing alone no longer suffices to create a distinctive customer value perception. Research has since expanded its scope to identify additional factors contributing to perceived value. Kotler (2008) extended the "perception" framework to include elements like "perceived profit" and "perception." This framework introduces new components such as personnel value and image value under "perceived value," while also considering non-monetary costs like time, energy, and psychological effort within the "perception" category.

Building on these research developments, and in the context of this study on agricultural insurance purchasing behavior, Sweeney and Soutar's (2001) model offers a useful framework. In this context, perceived value is divided into four key dimensions: functional value, emotional value, social value, and economic value.

## 2.4 Environmental concern

Environmental concern represents a positive attitude toward sustainability and reflects an individual's awareness of environmental issues and potential solutions [44]. These concerns encompass both specific environmental aspects, such as views on waste management, and broader considerations regarding humanity's relationship with the environment [45]. Environmentally conscious individuals tend to respond proactively to environmental challenges and are more likely to take actions that address them [46]. Numerous studies highlight the positive correlation between environmental concern and pro-environmental behaviors, such as energy conservation, recycling, and purchasing eco-friendly products [47,48]. This connection is often driven by the belief that green practices align with environmentally responsible production methods [49]. Furthermore, the strong association between environmental concern and individual moral obligations or personal norms is a critical driver of such behaviors [50].

Research by Tilikidou and Delistavrou [51] indicates that objective environmental knowledge has a direct impact on eco-conscious purchasing decisions. Similarly, Some studies found that consumers with higher environmental awareness are more likely to pay a premium for green products. Notably, previous studies also underscore the importance of subjective knowledge in shaping eco-friendly consumption patterns [52], emphasizing that perceived knowledge often plays a pivotal role in encouraging environmentally responsible behaviors. This growing focus on perceived knowledge suggests that consumers are increasingly prioritizing environmental considerations in their purchasing decisions.

In conclusion, this study underscores that consumers' subjective understanding of the environmental consequences of tea cultivation is a significant factor influencing their decision to purchase agricultural insurance. This finding aligns with the notion that perceived knowledge plays a key role in decision-making, even when objective knowledge may be limited. It highlights that individuals' confidence in their decisions is not always directly correlated with factual knowledge.

## 2.5 Purchase behavior

Building on Roger D. Blackwell's foundational research, the consumer behavior model has evolved into a comprehensive seven-step framework. This expanded model includes additional stages such as "post-purchase consumption" and "disposal," providing a more in-depth understanding of the purchasing process [53].

Purchase intention refers to an individual's subjective likelihood of carrying out a specific action, indicating their willingness to make a purchase. According to Ajzen, when actual control conditions are sufficient, purchase intention can directly influence actual purchasing behavior [54].

American scholar J.F. Engel categorized consumer purchasing behaviors into three types based on the effort consumers put into searching for and evaluating product information: extensive problem-solving, limited problem-solving, and routine/loyal purchasing behavior [55].

In the context of this study, purchase behavior is defined as the positive inclinations and concrete actions taken by tea farmers to obtain planting insurance. This process aims to convert their purchase intentions into real actions, considering factors such as government support and perceived value. The concept of purchasing behavior here highlights the practical steps that tea farmers take in their decision-making process, demonstrating how government interventions and perceived value shape their actions. Understanding this process provides valuable insights into the dynamics of agricultural insurance adoption and offers practical recommendations for policy development and increased farmer participation.

## 2.6 Conceptual framework

From the analysis of the literature review, and the consultation of the concepts, theories, and models regarding the relationship between variables, the following conceptual framework was developed.

Fig 1 shows that the conceptual framework of the article. The dependent variable is Purchase Behavior. In contrast, the independent variables include Government support and Risk cognition. And intervening variables include Perceived Value and Environmental concern.

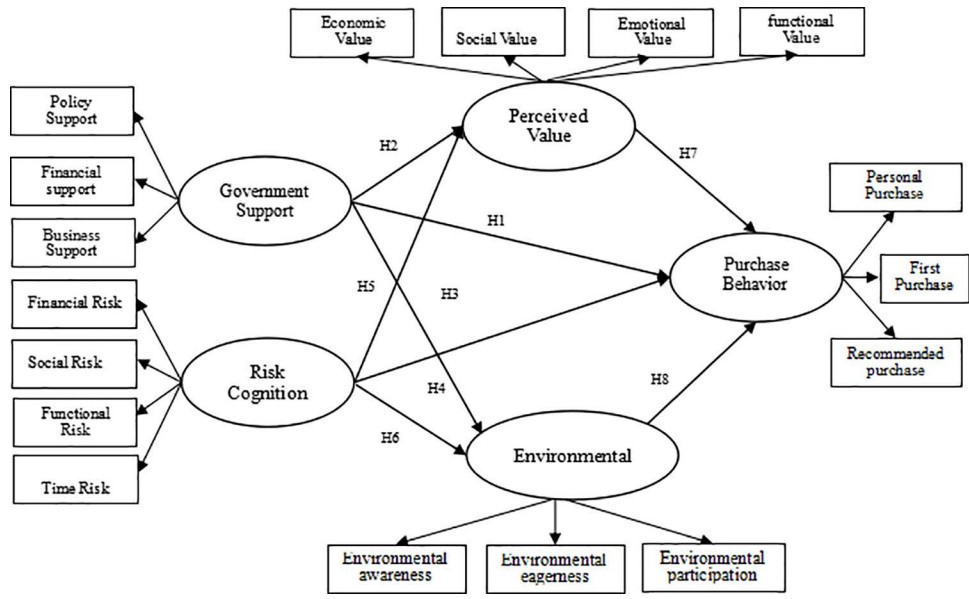

**Fig 1. Conceptual Framework of the study.**

As illustrated in Fig 1, the conceptual framework positions purchase behavior as the dependent variable, directly influenced by two independent variables: government support and risk cognition. Notably, this relationship is mediated through the intervening variables of perceived value and environmental concern, which collectively bridge the causal pathways between external factors and consumer decision-making.

Based on the synthesis of prior literature and the specific objectives of this study, the latent and observed variables presented in Table 1 were systematically identified.

## 3. Methodology

### 3.1 Population and sample

**3.1.1 Population.** The study focused on approximately 5,802 tea planters who are members of the Guizhou Tea Association, 2024. The research area was selected based on specific criteria to ensure the collection of relevant data on tea planters' purchasing behavior regarding planting insurance. Firstly, the study targeted primary tea-producing regions. Secondly, these areas had a well-established history of agricultural insurance development. Considering that Guizhou's tea garden area surpassed 7 million mu, with tea production reaching 501830 tons and a total output value of 96 billion yuan by the end of 2024 [58], the study specifically targeted representative tea farmers in Guizhou's tea-producing counties. A representative sample was drawn from this population to gather data on government support, risk perception, perceived value, environmental concern, and insurance purchasing behavior.

**3.1.2 Sample and sampling design and technique.** This study primarily explored the factors influencing the purchasing behavior of planting insurance among tea planters in Guizhou Province, China. The research targeted members of the Guizhou Tea Association. A random sampling approach was employed to ensure a representative selection from the population. The use of probability sampling aimed to create a sample that accurately reflects the population, giving each subject an equal chance of inclusion. Random samples are recognized for closely representing the population, thus enhancing the accuracy of the results [59]. A stratified random sampling technique was adopted to further ensure representativeness.

**Table 1. Latent, observed variables, and related literature.**

| Latent Variables | Observed Variables | Related Literature and Theory |
|---|---|---|
| Government support | -Policy support<br>-Financial support<br>-Business support | [14,15,16] |
| Risk cognition | -Financial risk cognition<br>-Social risk cognition<br>-Functional risk cognition<br>-Time risk cognition | [29,32,33] |
| Perceived Value | -Economic value<br>-Social value<br>-Emotional value<br>-Function value | [43,56,57] |
| Environmental concern | -Environmental awareness<br>-Environmental participation<br>-Environmental eagerness | [44,45,51] |
| Purchase Behavior | -Personal purchase<br>-Recommended purchase<br>-First purchase | [52,53,54] |

## 3.2 Reliability

To ensure the reliability and consistency of the data collected through the platform "Questionnaire Star," five experts were consulted to evaluate the validity of the survey instrument. The expert panel comprised two specialists in agricultural insurance and three professors with expertise in agriculture and business administration. A pilot study, involving 60 responses, was conducted to test the reliability of the questionnaire before the full survey. These pilot responses, along with their evaluations, were not included in the final analysis of the 550 questionnaires. Cronbach's Alpha (α) was employed to measure internal consistency, with Best and Kahn [60] suggesting that a value of 0.70 indicates acceptable reliability. In this study, the average Cronbach's Alpha was 0.988, reflecting a high degree of reliability [61,62].

The questionnaire was structured into two sections. The first section collected demographic data on the tea planters, while the second assessed their views on government support, risk perception, perceived value, environmental concern, and purchasing behavior. The second section comprised 51 items organized into five categories: government support (GS, 9 items), risk perception (RP, 12 items), perceived value (PV, 12 items), environmental concern (EC, 9 items), and purchasing behavior (PB, 9 items). Responses were rated on a 5-point Likert scale, ranging from 1 (strongly disagree) to 5 (strongly agree).

## 3.3 Statistical analysis

The validity of the questionnaire was assessed through a validity test to ensure that its items accurately measured the intended concepts. This involved testing aspects such as convergent validity, average variance extracted (AVE), and discriminant validity. Reliability was assessed using both Cronbach's Alpha and Composite Reliability [63]. A variable was considered reliable if it achieved a composite reliability score of 0.7, indicating a satisfactory level of consistency [64].

In PLS-SEM, Smart PLS 3.0 software was used to assess the reliability of constructs with reflective indicators through Cronbach's Alpha and Composite Reliability. A construct was deemed reliable if both its composite reliability and Cronbach's Alpha exceeded 0.70 [63]. The study applied both Cronbach's Alpha and composite reliability metrics to evaluate the reliability of the variables. According to Malhotra et al.[65], an item is considered reliable when its alpha coefficient exceeds 0.6.

## 3.4 Quantitative data analysis

Mertler [66] suggests that in educational research, a sample size of 300 is sufficient for a population of approximately 1,500, while for populations exceeding 5,000, a sample size of 400 is adequate. Increasing the sample size beyond this threshold yields diminishing returns, although a larger sample enhances the ability to generalize findings with greater confidence. In this study, 550 questionnaires were collected and reviewed, providing a sufficiently large sample to ensure reliability.

## 3.5 Confirmatory factor analysis (CFA)

A confirmatory factor analysis (CFA) was initially conducted to evaluate the measurement model, followed by structural equation modeling (SEM) to assess the model's fit and explore the relationships among constructs [67]. Wong [68] noted that, in marketing research, typical benchmarks include a 5% significance level, 80% statistical power, and $R^2$ values of at least 0.25. A model is considered acceptable when the p-value exceeds 0.05, and the $\chi^2$/df ratio is below 2 [69,70]. The root mean square error of approximation (RMSEA) is another commonly reported goodness-of-fit statistic, used to measure the fit of SEM models by evaluating the discrepancy per degree of freedom [71,72]. Hooper et al. (2008) recommended excluding items with multiple $R^2$ values of 0.20 or lower, as these indicate high error levels. Hair et al. (2016) proposed that $R^2$ values should exceed 0.25 to meet more stringent criteria.

### 3.6 Ethics statement

This study was approved by the Ethics Committee of Tongren Polytechnic college (Approval time: July 6, 2024). The recruitment period for this study began on July 7, 2024 and ended on September 16, 2024. Written informed consent was obtained from all participants, who were fully informed about the study prior to participation. All data were anonymized to ensure privacy and were used solely for the purposes of this research.

## 4. Empirical results

### 4.1 Respondents' characteristics (n = 550)

Table 2 presents the demographic characteristics of the survey respondents, providing an overview of the sample distribution. In terms of gender, the majority of respondents are male, accounting for 71.27% of the total sample. Regarding age, 79.64% of the respondents are over 36 years old. As for the scale of tea planting, 25.82% of respondents manage less than 5 acres, 28.55% manage between 6–10 acres, and only 7.64% manage more than 21 acres. In terms of years of experience, 51.81% of the respondents have been engaged in tea planting for more than 7 years. With respect to education, 68.36% of the respondents have completed junior middle school, while only 2.18% have attained a bachelor's degree or higher.

### 4.2 Respondents' information

Table 3 presents the mean and standard deviation (S.D.) for each variable measured in the study. All variables have mean scores exceeding 3.50, indicating a high level of agreement or positive perception among respondents. The standard deviations, ranging from 0.40616 to 0.45248, suggest consistent perceptions across respondents for the measured constructs.

**Table 2. Demographic Information (n = 550).**

| Variable | Items | Frequency | Percent |
|---|---|---|---|
| Gender | Male | 392 | 71.27 |
| | Female | 158 | 28.73 |
| Age | Less than 25 years old | 32 | 5.82 |
| | 26-35 years old | 80 | 14.55 |
| | 36-45 years old | 286 | 51.99 |
| | More than 46 years old | 152 | 27.64 |
| Planting scale | Less than 5 acres | 142 | 25.82 |
| | 6-10 acres | 157 | 28.54 |
| | 11-15 acres | 63 | 11.45 |
| | 16-20 acres | 146 | 26.55 |
| | More than 21 acres | 42 | 7.64 |
| Planting years | Less than 3 years | 102 | 18.55 |
| | 4-6 years | 163 | 29.64 |
| | 7-9 years | 145 | 26.36 |
| | More than 10 years | 140 | 25.45 |
| Education | Junior middle school | 376 | 68.37 |
| | Senior middle school | 106 | 19.27 |
| | Junior college | 56 | 10.18 |
| | Bachelor's degree or above | 12 | 2.18 |

**Table 3. Mean and standard deviation for each variable.**

| Variable | Mean | S.D. |
|---|---|---|
| Government support (IGS) | 3.6166 | .42141 |
| Risk cognition (IRC) | 4.0260 | .43548 |
| Perceived value (MPV) | 3.9004 | .40616 |
| Environmental concern (MEC) | 4.0353 | .45248 |
| Purchase behavior (DPB) | 4.3511 | .40545 |

Following a comprehensive review of relevant literature and theoretical frameworks, a confirmatory factor analysis (CFA) was conducted to examine the relationships between internal and external variables. Using AMOS 28.0 to analyze the CFA results, the model was found to be statistically non-significant (P>0.05), with a $\chi^2$/df ratio ≤ 2.00 and an RMSEA value ≤ 0.05. The comparative fit index (CFI) was reported at 0.986, indicating strong model fit, as CFI and Tucker-Lewis Index (TLI) values above 0.90 are generally considered acceptable, with values exceeding 0.95 being ideal [73]. Additionally, the adjusted goodness-of-fit index (AGFI) was recorded at 0.982, further demonstrating a strong model fit, as AGFI values above 0.90 indicate good fit.

Fig 2 presents the results of the CFA (confirmatory factor analysis) conducted on the external latent variables IGS (Independent variable Government Support) and IRC (Independent variable Risk Cognition) with a sample size of 550.

The Fig 2 presents the results of the CFA (confirmatory factor analysis) conducted on the external latent variables IGS (Independent variable Government Support) and IRC (Independent variable Risk Cognition) with a sample size of 550. The model fit indices are as follows: Chi-Square ($\chi^2$) = 8.439 with 14 degrees of freedom (df), Comparative Fit Index (CFI) = 0.981, Tucker-Lewis Index (TLI) = 0.960, and Root Mean Square Error of Approximation (RMSEA) = 0.026. These indices suggest a good fit of the model to the data.

Note: Chi-Square ($\chi^2$)= 8.439, df = 14, CFI=0.981,TLI=0.960, RMSEA = 0.026.

Fig 3 shows the results of the confirmatory factor analysis (CFA) on the external latent variables MPV (Mediating variable Perceived Value), MEC (Mediating variable Environmental Concern), and DPB(Dependent variable Purchase Behavior) with a sample size of 550.

Fig 3 shows the results of the confirmatory factor analysis (CFA) on the external latent variables MPV (Mediating variable Perceived Value), MEC (Mediating variable Environmental Concern), and DPB(Dependent variable Purchase Behavior) with a sample size of 550. Model fit indices are as follows: Chi-Square ($\chi^2$) = 34.872 with 32 degrees of freedom (df),

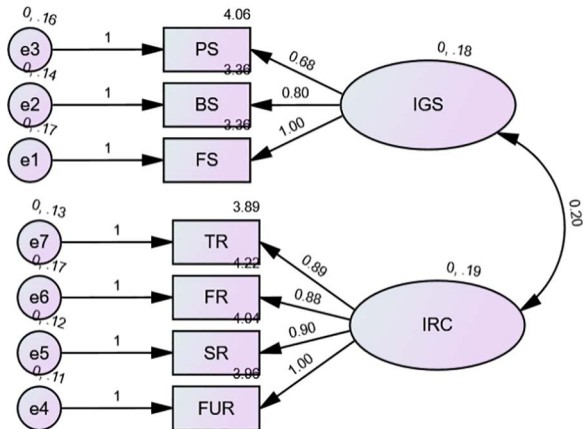

**Fig 2. CFA of external latent variables IGS & IRC (n = 550).**

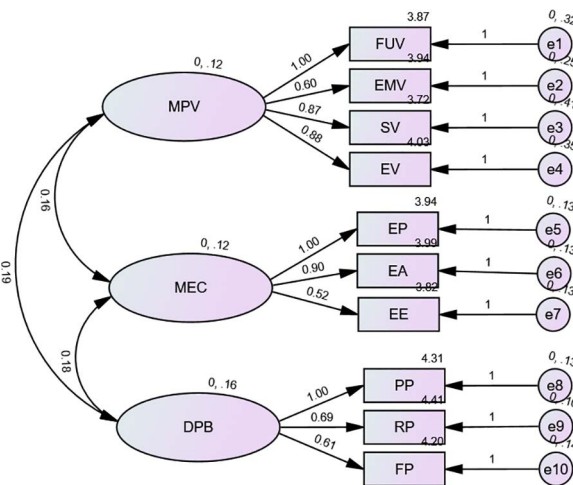

**Fig 3. CFA of external latent variables MPV, MEC & DPB (n = 550).**

Comparative Fit Index (CFI) = 0.933, Tucker-Lewis Index (TLI) = 0.942, and Root Mean Square Error of Approximation (RMSEA) = 0.024. These indices indicate an acceptable fit of the model to the data.

Note: Chi-Square($\chi^2$)= 34.872, df = 32, CFI=0.933, TLI=0.942, RMSEA = 0.024.

### 4.3 Convergent model analysis

The analysis of the data, alongside the evaluation of the five constructs and their related hypotheses, revealed a strong model fit with the empirical research findings. Both convergent and discriminant validity were assessed to ensure accuracy. In structural equation modeling (SEM), confirmatory factor analysis (CFA) is typically employed to evaluate construct validity [74]. Hair et al. (2016) and Byrne et al. (2021) emphasized that factor loadings or regression weights for latent variables to observed variables should exceed 0.50, indicating that all constructs meet the criteria for construct validity and exhibit appropriate convergence.

The results presented in Table 4 show a $\chi^2$ value of 139.930 with a df ratio of 1.284, demonstrating statistical significance as it exceeded 0.05. This supports the hypothesis that the model aligns with the empirical data. Further validation was provided by the goodness-of-fit index (GFI), which was 0.962, and the adjusted goodness-of-fit index (AGFI), recorded at 0.981 [75]. Additionally, the root mean square error of approximation (RMSEA) was 0.013. As RMSEA is an absolute fit measure, a value of zero reflects a perfect fit, and values below 0.05 indicate a good fit [72].

AVE (Average Variance Extracted) is a metric used to evaluate the convergent validity of Latent Variable in a measurement model. It measures the ability of latent variables to interpret variance of their measures (explicit variables), reflecting the validity and reliability of the measurement model. AVE ≥ 0.50 indicates good convergence validity of the measurement model. Table 5 summarizes the interrelationships and reliability assessments among the variables. Key indicators, including composite reliability (CR > 0.8), and average variance extracted (AVE > 0.6), align with established psychometric standards. These results confirm the dataset's suitability for subsequent analyses, such as structural equation modeling (SEM) or mediation effect testing.

As detailed in Table 6, the hypothesis testing framework assessed statistical significance (α = 0.05), standardized path coefficients (β), and critical ratios (CR) with associated standard errors (SE) to empirically validate each hypothesis.

**Table 4. Criteria and theory of the goodness-of-fit values.**

| Criteria Index | Criteria | Values | Results | Supporting theory |
|---|---|---|---|---|
| Chi-square: $\chi^2$ | $p \geq 0.05$ | 139.930 | passed | Measure whether the distribution of categorical variables corresponds to the desired theoretical distribution [75]. |
| $\chi^2$ | $\leq 2.00$ | 1.284 | passed | Measure how well the theoretical model fits the actual data./df ≤ 2.00, the model fit is very good [69]. |
| GFI | $\geq 0.90$ | 0.962 | passed | GFI(Goodness-of-Fit Index) measure how well the theoretical model fits the actual data. GFI > 0.90 indicates that the model fits well [63,74]. |
| AGFI | $\geq 0.90$ | 0.981 | passed | AGFI (Adjusted Goodness-of-Fit Index) is an adjusted version of GFI that is used to evaluate the goodness-of-fit of structural equation models. AGFI introduces a degree of freedom adjustment on top of GFI, which can more reasonably reflect the complexity and fitting effect of the model. AGFI > 0.90 indicates that the model fits well.[75]. |
| RMSEA | $\leq 0.05$ | 0.013 | passed | RMSEA (Root Mean Square Error of Approximation) is an index used to evaluate the goodness of fit of Structural Equation Model (SEM). It measures the approximate error between the model and the data, reflecting how well the model fits in the population. An RMSEA < 0.05 indicates that the model fits very well [72]. |
| Cronbach's Alpha | $\geq 0.70$ | 0.896 | passed | Cronbach's Alpha is a statistical measure used to assess the reliability of the internal consistency of a scale or questionnaire. It measures the correlation between the items in the scale, reflecting the reliability and stability of the scale. α≥0.70 indicates that the scale reliability is relatively good [76]. |

## 4.4 Mediator effect analysis

The mediation effect of the model was analyzed by AMOS28.0, with detailed bootstrap results presented in Table 7. If the significant value of the mediation effect is less than 0.05, the results are credible. According to the values in the table, the two mediation variables, perceived value and environmental concern, are partial mediators.
So, the following assumptions hold.

> H9: Perceived value mediates the relationship between government support and tea planters' planting insurance purchasing behavior. Government support, such as educational initiatives and technical services, has enhanced tea planters' trust and recognition of insurance, making them more aware of its practical benefits and encouraging them to purchase insurance more actively. Tea planters who perceive higher value from insurance believe that the impact of government support is more significant.

**Table 5. The correlation coefficient, reliability, and AVE of the latent variables.**

| Latent variables | IGS | IRC | MPV | MEC | DPB |
|---|---|---|---|---|---|
| IGS | 1.00 | | | | |
| IRC | .830** | 1.00 | | | |
| MPV | .755** | .717** | 1.00 | | |
| MEC | .852** | .816** | .689** | 1.00 | |
| DPB | .827** | .853** | .706** | .901** | 1.00 |
| Construct Reliability | 0.875 | 0.902 | 0.872 | 0.884 | 0.910 |
| AVE | 0.701 | 0.844 | 0.723 | 0.693 | 0.616 |
| $\sqrt{AVE}$ | 0.837 | 0.919 | 0.850 | 0.833 | 0.785 |

Note: **. At the 0.01 level (two-tailed), the correlation was significant.

**Table 6. Hypotheses testing results.**

| Hypotheses | Estimate | S.E. | C.R. | P-Label | Results |
|---|---|---|---|---|---|
| H1: Government support has a positive impact on tea planters' planting insurance purchasing behavior. (DPB<--IGS) | 0.181 | 0.027 | 6.774 | *** | Accepted |
| H2: Government support has a positive impact on the perceived value of tea planters. (MPV<--IGS) | 1.539 | 0.218 | 7.053 | *** | Accepted |
| H3: Government support has a positive impact on tea planters' environmental concern. (MEC<--IGS) | 0.306 | 0.033 | 9.4 | *** | Accepted |
| H4: Risk cognition has a positive impact on tea planters' planting insurance purchasing behavior. (DPB<--IRC) | 0.231 | 0.028 | 8.242 | *** | Accepted |
| H5: Risk cognition has a positive impact on the perceived value of tea planters. (MPV<--IRC) | 0.285 | 0.037 | 7.737 | *** | Accepted |
| H6: Risk cognition has a positive impact on tea planters' environmental concern. (MEC<--IRC) | 0.153 | 0.03 | 5.11 | *** | Accepted |
| H7: Perceived value has a positive impact on tea planters' planting insurance purchasing behavior. (DPB<--MPV) | 0.222 | 0.046 | 4.839 | *** | Accepted |
| H8: Environmental concern has a positive impact on tea planters' planting insurance purchasing behavior. (DPB<--MEC) | 0.24 | 0.095 | 2.526 | 0.001 | Accepted |
| H13: Perceived value has a positive impact on tea planters' environmental concern. (MEC<--MPV) | 0.489 | 0.048 | 10.243 | *** | Accepted |

Note: *Sig.<0.05, **Sig.<0.01, ***Sig.<0.001, Critical ratios (t-values) more than 1.96 are significant at the 0.05 level. S.E.=standard error, CR=critical ratio(t-value).

**Table 7. Results of the mediation effect test.**

| | Parameter | Estimate | Lower | Upper | P |
|---|---|---|---|---|---|
| IGS→MPV→DPB | mediation effect | .237 | .185 | .288 | .009 |
| | total effect | .296 | .191 | .401 | .009 |
| | ratio | .800 | .615 | 1.160 | .008 |
| IGS→MEC→DPB | mediation effect | .173 | .123 | .241 | .007 |
| | total effect | .232 | .139 | .334 | .007 |
| | ratio | .746 | .516 | 1.225 | .007 |
| IRC→MPV→DPB | mediation effect | .072 | .041 | .126 | .002 |
| | total effect | .175 | .105 | .293 | .002 |
| | ratio | .410 | .236 | .727 | .007 |
| IRC→MEC→DPB | mediation effect | .069 | .031 | .114 | .013 |
| | total effect | .172 | .108 | .261 | .009 |
| | ratio | .400 | .158 | .629 | .028 |

H10: Perceived value mediates the relationship between risk cognition and tea planters' planting insurance purchasing behavior. In the tea planting process, planters assess the balance between the benefits and costs of purchasing planting insurance. The higher their perceived value, the more willing they are to invest in insurance. However, their purchasing decision is influenced by risk perception, with perceived value acting as a mediator between risk perception and the intention to purchase planting insurance.

H11: Environmental concern mediates the relationship between government support and tea planters' planting insurance purchasing behavior. Moreover, government support stimulates tea planters' environmental awareness through

subsidies for environmental protection and the promotion of eco-friendly agricultural technologies. This, in turn, increases their attention to environmental protection. As their awareness of ecological risks grows, including the impact of climate change and ecological degradation, they become more likely to purchase planting insurance as a precautionary measure to mitigate these risks.

H12: Environmental concern mediates the relationship between risk cognition and tea planters' planting insurance purchasing behavior. Tea planters with high environmental concerns are more attuned to the risks posed by environmental factors, which strengthens their awareness of the uncertainties and potential hazards associated with tea planting, prompting them to take proactive steps to avoid such risks.

## 4.5 Structural Equation Modeling (SEM) results

Hooper et al. (2008) suggested that items with low $R^2$ values ($\leq 0.20$) should be excluded from analysis, as they indicate high levels of error. The results of the structural equation modeling (SEM), presented in the study (Fig 4), confirm that the model meets the necessary criteria, with the index found to be non-significant. Furthermore, all causal factors in the model demonstrated a positive impact on purchasing behavior (DPB). Among these, environmental concern (MEC) exerted the strongest influence on purchasing behavior, with a coefficient of 0.24, followed by risk perception (IRC) at 0.23, and perceived value (MPV) at 0.22.

Fig 4 presents the final structural equation model (SEM) with estimated values based on a sample size of 550. Model fit statistics are as follows: Chi-Square ($\chi^2$) = 139.930 with 109 degrees of freedom (df), p-value = 0.64514, and Root Mean Square Error of Approximation (RMSEA) = 0.013. These values indicate a strong fit of the model to the observed data, suggesting that the final SEM adequately represents the relationships among the variables.

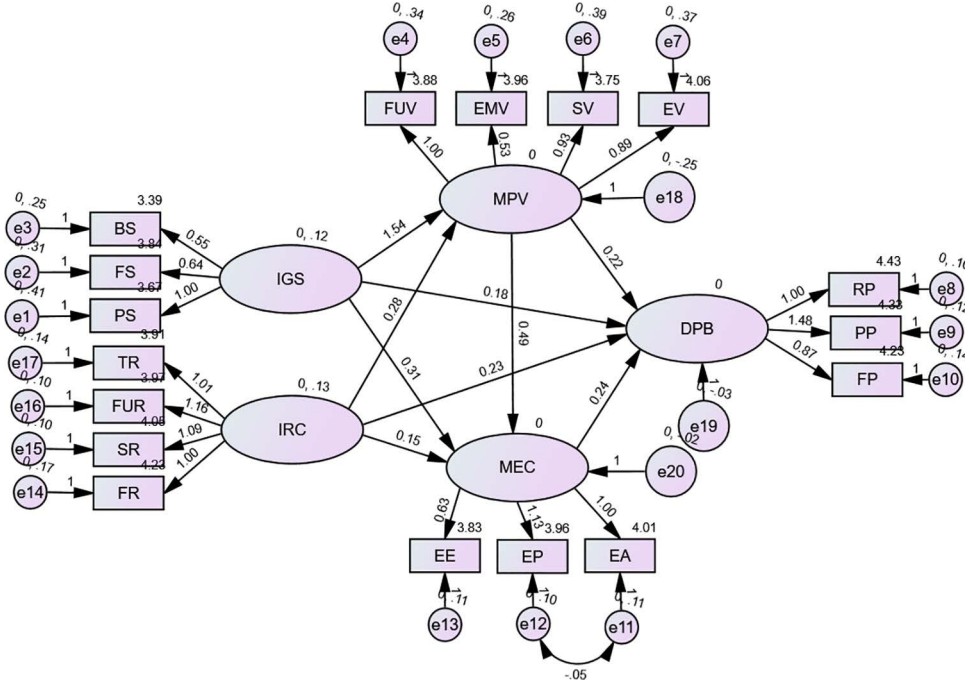

**Fig 4. SEM final model with values from estimates (n = 550).**

Note: Chi-Square ($\chi^2$)=139.930, df=109, P-value=0.64514, RMSEA=0.013

## 5. Discussion

The results of this study provide strong support for "H1", demonstrating that government support (IGS) has a positive impact on tea planters' planting insurance purchasing behavior (DPB), with a correlation coefficient of 0.18. This conclusion is consistent with the responses of tea planters interviewed in the survey, who believe that government support can help reduce the economic losses of tea planters in terms of tea insurance types, premiums and claims. This finding aligns with research by Xiao et al. (2013), which shows that increasing government premium subsidies is positively correlated with farmers' willingness to engage in insurance. Similar conclusions were drawn by Mahul & Stutley (2010) and Kousar (2023), who found that heightened government subsidies significantly enhance farmer participation in agricultural insurance[77,78,79].

"H2" was also supported, showing that government support (IGS) positively influences the perceived value (MPV) of tea planters, with a coefficient of 1.54. In the interview, tea planters said that the government's knowledge of tea insurance can improve tea planters' awareness of the function of tea insurance and improve the perceived value of tea planters. When government support is aligned with public expectations and aims to enhance satisfaction, the perceived value and trust in government initiatives increase [80].

"H3" was supported as well, confirming that government support has a positive effect on tea planters' environmental concern, with a coefficient of 0.31. In interviews, tea planters said that the government has helped tea planters reduce production risks by providing services such as weather warnings and pest control, which indirectly increases their demand for insurance. By educating tea planters about the impact of climate change and the role of insurance, the government raises awareness of environmental risks among tea planters, thereby encouraging them to purchase insurance. Tea planters with high environmental concerns are more receptive to the government's insurance promotion policies.

The results also validate "H4", indicating that risk cognition has a positive impact on tea planters' planting insurance purchasing behavior, with a coefficient of 0.23. Tea planters in this interview agree that when they are aware of the risks they may face during the planting process (e.g., drought, floods, pests and diseases, etc.), they are more inclined to buy insurance to protect against potential losses.This finding is consistent with previous research, which emphasizes the role of risk awareness in influencing farmers' participation in insurance [81].

"H5" was supported, showing that risk cognition positively affects the perceived value of tea planters, with a coefficient of 0.28. Studies conducted in South Africa, Sweden, and the USA also confirm the connection between motivational values and risk perception dimensions [82].

"H6" was supported, demonstrating that risk cognition positively influences tea planters' environmental concern, with a coefficient of 0.15. In the interview, tea planters said that a high level of risk cognition will motivate tea farmers to participate in environmental protection activities, such as soil and water conservation, ecological restoration, etc., to reduce the impact of environmental risks on tea plantations.

"H7" was supported, highlighting the positive impact of perceived value on tea planters' planting insurance purchasing behavior, with a coefficient of 0.22. This result supports previous findings, which emphasize that perceived value directly influences purchasing behavior [83].

"H8" was supported, confirming that environmental concern has a positive impact on tea planters' planting insurance purchasing behavior, with a coefficient of 0.24.

"H9" was supported, revealing that perceived value mediates the impact of government support on tea planters' planting insurance purchasing behavior, with a mediation effect value of 0.237. This aligns with Zhang [14], who demonstrated the mediating role of premium subsidies in influencing rice farmers' insurance participation in Sichuan Province.

"H10" was supported, showing that perceived value mediates the impact of risk cognition on tea planters' planting insurance purchasing behavior, with a mediation effect value of 0.072.

"H11" was supported, indicating that environmental concern mediates the impact of government support on tea planters' planting insurance purchasing behavior, with a mediation effect value of 0.173. This is consistent with research suggesting that individuals who value environmental issues tend to evaluate the environmental consequences of their purchasing decisions [84].

"H12" was also supported, revealing that environmental concern mediates the impact of risk cognition on tea planters' planting insurance purchasing behavior, with a mediation effect value of 0.069.

## 6. Conclusion

This study provides a comprehensive examination of the factors influencing tea growers' decisions to purchase planting insurance, with a particular focus on the effects of government support, risk cognition, and perceived value. Utilizing a conceptual framework that designates purchasing behavior as the outcome variable, with government support and risk cognition as antecedents and perceived value as a mediator, several hypotheses were formulated and empirically tested.

The analysis confirms that both government support and risk cognition significantly impact perceived value and the adoption of planting insurance. Moreover, perceived value was found to foster positive purchasing behavior among tea planters. These findings highlight the importance of considering multiple factors—such as external support systems, individuals' risk assessments, and subjective value judgments—when understanding and influencing consumer behavior in the agricultural insurance sector.

These insights offer valuable implications for policymakers, insurance providers, and agricultural stakeholders, enabling them to design targeted interventions that promote insurance adoption and reduce risks for tea growers. The application of heterogeneity testing methods enables the precise identification of diverse needs across different categories of tea planters, consequently improving their awareness and adoption of crop insurance. Specifically, small-scale planters tend to emphasize affordability of premiums and ease of enrollment. To cater to these priorities, policy interventions could include enhanced premium subsidy schemes, the introduction of micro-insurance options, and streamlined enrollment processes. These measures would effectively lower participation thresholds and mitigate financial pressures. Conversely, large-scale planters typically demand tailored insurance solutions and comprehensive risk management support. This can be achieved through the development of sophisticated insurance products featuring extended coverage and multi-risk protection, complemented by professional risk management advisory services, thereby optimizing their insurance portfolios and elevating protection standards.Moreover, the creation of an integrated database encompassing tea planter information would serve as a critical foundation for evidence-based policy development. Implementing pilot initiatives coupled with adaptive policy refinement allows for continuous improvement of intervention strategies. It is imperative to foster synergistic partnerships among government entities, insurance providers, academic institutions, and agricultural cooperatives to collaboratively design and execute these policies. Such multi-stakeholder engagement ensures not only effective policy implementation but also sustained optimization to address the dynamic and varied requirements of tea planters. Looking forward, future research could explore additional variables and refine the conceptual framework to further enhance the understanding of agricultural insurance and consumer behavior dynamics.

## Author contributions

**Conceptualization:** Zhenhua Xu, Nuttawut Rojniruttikul.

**Data curation:** Zhenhua Xu, Nuttawut Rojniruttikul.

**Formal analysis:** Zhenhua Xu.

**Methodology:** Zhenhua Xu, Nuttawut Rojniruttikul.

**Resources:** Zhenhua Xu.

**Writing – original draft:** Zhenhua Xu, Nuttawut Rojniruttikul.

**Writing – review & editing:** Zhenhua Xu, Nuttawut Rojniruttikul.

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
