## [Decision Letter · Decision Letter 0]

24 Jan 2025

PONE-D-24-48851Determinants of Tea Planters' Purchasing Behavior of Planting Insurance: SEM analysisPLOS ONE

Dear Dr. Rojniruttikul,

Thank you for submitting your manuscript to PLOS ONE. After careful consideration, we feel that it has merit but does not fully meet PLOS ONE’s publication criteria as it currently stands. Therefore, we invite you to submit a revised version of the manuscript that addresses the points raised during the review process.

We look forward to receiving your revised manuscript.

Kind regards,

Xufeng Cui, Ph.D

Academic Editor

PLOS ONE

Reviewers' comments:

Reviewer's Responses to Questions

**Comments to the Author**

1. Is the manuscript technically sound, and do the data support the conclusions?

Reviewer #1: Yes

Reviewer #2: Yes

2. Has the statistical analysis been performed appropriately and rigorously? 

Reviewer #1: Yes

Reviewer #2: Yes

3. Have the authors made all data underlying the findings in their manuscript fully available?

Reviewer #1: Yes

Reviewer #2: Yes

4. Is the manuscript presented in an intelligible fashion and written in standard English?

Reviewer #1: Yes

Reviewer #2: Yes

5. Review Comments to the Author

Reviewer #1: Dear Author,

The researchers stated that this study explores the influence of government support and risk cognition on tea planters' purchasing behavior regarding planting insurance, with a particular emphasis on the mediating roles of perceived value and environmental concern. Data were collected from 550 tea planters in Guizhou Province, China, using a structured questionnaire and convenience sampling method. Confirmatory factor analysis (CFA) and structural equation modeling (SEM) were conducted using AMOS 28.0 to analyze the data. The results indicate that both government support and risk cognition significantly and positively impact the perceived value of planting insurance. To promote the uptake of planting insurance among tea farmers, government agencies should strengthen policy advocacy and provide business guidance. Such efforts would help tea farmers recognize the value and psychological significance of planting insurance, thereby better safeguarding their agricultural interests. This study enhances the understanding of how government support and risk cognition can facilitate the adoption of planting insurance among tea farmers. However, the research paper demonstrates low level of understanding of the relevant literature in the field and did not cite an appropriate range of literature sources. Methodology is good. Analyses and findings are presented in a weak manner as to present new ideas. Also, the research has not proper discussion and conclusion. Though, the paper needs improvements in order to meet the standards of this journal. Also:

• It’s better to update some of the sources such as ITC, 2022; The China Tea Circulation Association, 2023; Zhang & Fan, 2016; Guizhou Tea Association, 2022 into recent sources such as 2024.

• The title 3.4 Qualitative data analysis should be modified as the research did not tackle qualitative data analysis.

• Table 2. Demographic Information (n =550) should contain the totals for each variable. Also, the sum of variables percentages should be 100% not 99.9%.

• The results in Table 6. Hypotheses testing results should be modified into Accepted instead of consistent.

Finally, the references should be ordered alphabetically and unify the writing style based on the journal’s layout.

Reviewer #2: Comments and Suggestions for Authors

I would like to express my appreciation to the authors for their efforts in producing this manuscript. I found it engaging and informative, but there are a few aspects that could be improved to enhance its clarity and impact.

First of all, regarding the research methods employed in this article, qualitative and quantitative methods such as questionnaires and structural equation modeling have been utilized, and their application and analysis are relatively reasonable. However, it is advisable to consider supplementing with additional qualitative research, such as individual interviews, to gain a more comprehensive understanding of the motives behind tea farmers' purchasing behavior.

Secondly, in the data analysis section of the article, the mediating role of perceived value and environmental concern in tea farmers' decision to purchase planting insurance is emphasized. Yet, further exploration is needed into the relationship between perceived value and environmental concern, as well as how these two factors influence risk perception and policy support.

Furthermore, the article lists several model fit indices (such as χ²/df, RMSEA, CFI, TLI, AGFI, etc.), but does not provide sufficient explanation for the specific meanings of these indices. Detailed explanations of these model fit indices can be added in the results or notes section to assist readers in understanding the fit of the models.

Lastly, in the policy recommendations section, the article mentions some targeted policy suggestions, such as increasing government subsidies and providing business guidance, which are helpful in enhancing tea farmers' awareness and acceptance of planting insurance. In addition to these, further discussion is needed on how to design more targeted policy measures, for instance, by adopting heterogeneity testing methods to better meet the needs of different types of tea farmers (considering factors such as operating scale, planting history, geographical location, etc.).

I hope these comments and suggestions are helpful, and I look forward to seeing the revised version.

6. PLOS authors have the option to publish the peer review history of their article (what does this mean? ). If published, this will include your full peer review and any attached files.

**Do you want your identity to be public for this peer review?** For information about this choice, including consent withdrawal, please see our Privacy Policy .

Reviewer #1: No

Reviewer #2: No

---

## [Author Response · Author response to Decision Letter 0]

19 Feb 2025

Thank you for your careful consideration of our manuscript. Your feedback has significantly improved the overall quality of the paper, and we are grateful for your contributions.

We sincerely appreciate your time and expertise in reviewing our manuscript. Your suggestions have led to meaningful improvements in the clarity and coherence of the paper.

The following text is our response to the reviewers' comments:

Reviewer #1:

1.It’s better to update some of the sources such as ITC, 2022; The China Tea Circulation Association, 2023; Zhang & Fan, 2016; Guizhou Tea Association, 2022 into recent sources such as 2024.

Regarding P1-P3 and P11:

We have updated the data sources cited in the manuscript. Specifically, we replaced the data with the latest available figures from the World Tea Association (2023) and the China Tea Circulation Association (2024), as the 2024 data from the World Tea Association are not yet available.

Additionally, the reference to Zhang & Fan (2016) has been updated to the Insurance Association of China's "China Agricultural Insurance Market Demand Survey Report (2022)" to ensure the use of the most recent and relevant data.

Regarding P11: The data from the Guizhou Tea Association has been updated from 2022 to the most recent 2024 data to reflect the latest trends and statistics.

2.The title 3.4 Qualitative data analysis should be modified as the research did not tackle qualitative data analysis.

Regarding P13: As suggested, we have revised the qualitative data analysis section to quantitative data analysis to align with the methodological focus of the study and provide a more robust analytical framework.

3.Table 2. Demographic Information (n =550) should contain the totals for each variable. Also, the sum of variables percentages should be 100% not 99.9%.

Regarding P14 (Table 2): We have adjusted the percentages in Table 2 using the "maximum residual method". Specifically:

If the sum of percentages is 100.01%, the largest item is reduced by 0.01%.

If the sum is 99.99%, the largest item is increased by 0.01%.

This adjustment ensures that the total percentage sums to exactly 100%, addressing the rounding discrepancies noted in the original manuscript.

4.The results in Table 6. Hypotheses testing results should be modified into Accepted instead of consistent.

Regarding P18 (Table 6): In the Hypotheses Testing Results section, we have revised the terminology to clearly indicate that the hypotheses have been "Accepted" based on the statistical analysis. This change aligns with the standard academic convention for reporting hypothesis testing outcomes.

Reviewer #2:

First of all, regarding the research methods employed in this article, qualitative and quantitative methods such as questionnaires and structural equation modeling have been utilized, and their application and analysis are relatively reasonable. However, it is advisable to consider supplementing with additional qualitative research, such as individual interviews, to gain a more comprehensive understanding of the motives behind tea farmers' purchasing behavior.

Regarding the hypothesis testing and data analysis (P21-P22): We have enhanced our hypothesis validation by incorporating qualitative insights obtained through direct communication and interviews with tea farmers during the survey process. These field observations serve as valuable supplementary evidence to our quantitative data analysis, providing a more comprehensive understanding of tea planters' perspectives and behaviors.

Secondly, in the data analysis section of the article, the mediating role of perceived value and environmental concern in tea farmers' decision to purchase planting insurance is emphasized. Yet, further exploration is needed into the relationship between perceived value and environmental concern, as well as how these two factors influence risk perception and policy support.

Concerning the discussion of perceived value and environmental concern (P20-P21): In response to H13 (Perceived value has a positive impact on tea planters' environmental concern), we have expanded our analysis in P21 to examine the mediating relationships between perceived value, environmental concern, government support, and risk perception. Furthermore, we have thoroughly investigated how these factors collectively influence the purchase behavior of tea planting insurance, providing deeper insights into the decision-making process of tea planters.

Furthermore, the article lists several model fit indices (such as χ²/df, RMSEA, CFI, TLI, AGFI, etc.), but does not provide sufficient explanation for the specific meanings of these indices. Detailed explanations of these model fit indices can be added in the results or notes section to assist readers in understanding the fit of the models.

Regarding the model fit indices and measurement evaluation (P16-P18):

We have enhanced the explanatory notes in Table 4 to provide clearer interpretation of the model fit indices, including χ²/df, RMSEA, CFI, TLI, and AGFI. Additionally, in Table 5, we have included a more detailed explanation of the Average Variance Extracted (AVE) alongside the correlation coefficients and reliability measures, ensuring better understanding of our measurement model's validity and reliability.

Lastly, in the policy recommendations section, the article mentions some targeted policy suggestions, such as increasing government subsidies and providing business guidance, which are helpful in enhancing tea farmers' awareness and acceptance of planting insurance. In addition to these, further discussion is needed on how to design more targeted policy measures, for instance, by adopting heterogeneity testing methods to better meet the needs of different types of tea farmers (considering factors such as operating scale, planting history, geographical location, etc.).

Concerning policy implications and heterogeneity analysis (P24-P25):

We have incorporated a comprehensive discussion on designing targeted policy measures through heterogeneity detection methods. Our analysis reveals that:

Small-scale planters primarily focus on premium affordability and enrollment convenience. To address these needs, we recommend implementing higher premium subsidy ratios, developing micro-insurance products, and streamlining enrollment procedures.

Large-scale planters require more sophisticated solutions, including customized insurance products and comprehensive risk management services. We suggest developing insurance products with higher coverage limits and multiple risk protections, complemented by professional risk management consulting services.

To support these policy recommendations, we propose establishing a comprehensive tea planter information database, implementing pilot programs for policy testing, and fostering collaboration among government agencies, insurance companies, research institutions, and cooperatives. This multi-stakeholder approach will ensure continuous policy refinement and effective implementation, ultimately better serving the diverse needs of tea planters.

We would like to express our sincere gratitude to the reviewers for their valuable comments and constructive suggestions, which have significantly improved the quality of our manuscript. We appreciate the time and effort dedicated to reviewing our work, and we believe these revisions have strengthened the paper's overall contribution.

---

## [Decision Letter · Decision Letter 1]

28 Mar 2025

Determinants of Tea Planters' Purchasing Behavior of Planting Insurance: SEM analysis

PONE-D-24-48851R1

Dear Dr. Rojniruttikul,

We’re pleased to inform you that your manuscript has been judged scientifically suitable for publication and will be formally accepted for publication once it meets all outstanding technical requirements.

Kind regards,

Xufeng Cui, Ph.D

Academic Editor

PLOS ONE

Additional Editor Comments (optional):

Reviewers' comments:

Reviewer's Responses to Questions

**Comments to the Author**

1. If the authors have adequately addressed your comments raised in a previous round of review and you feel that this manuscript is now acceptable for publication, you may indicate that here to bypass the “Comments to the Author” section, enter your conflict of interest statement in the “Confidential to Editor” section, and submit your "Accept" recommendation.

Reviewer #1: All comments have been addressed

Reviewer #2: (No Response)

2. Is the manuscript technically sound, and do the data support the conclusions?

Reviewer #1: Yes

Reviewer #2: (No Response)

3. Has the statistical analysis been performed appropriately and rigorously? 

Reviewer #1: Yes

Reviewer #2: (No Response)

4. Have the authors made all data underlying the findings in their manuscript fully available?

Reviewer #1: Yes

Reviewer #2: (No Response)

5. Is the manuscript presented in an intelligible fashion and written in standard English?

Reviewer #1: Yes

Reviewer #2: (No Response)

6. Review Comments to the Author

Reviewer #1: Dear Author,

Since the researchers stated that this study explores the influence of government support and risk cognition on tea planters' purchasing behavior regarding planting insurance, with a particular emphasis on the mediating roles of perceived value and environmental concern. Data were collected from 550 tea planters in Guizhou Province, China, using a structured questionnaire and convenience sampling method. Confirmatory factor analysis (CFA) and structural equation modeling (SEM) were conducted using AMOS 28.0 to analyze the data. The results indicate that both government support and risk cognition significantly and positively impact the perceived value of planting insurance. To promote the uptake of planting insurance among tea farmers, government agencies should strengthen policy advocacy and provide business guidance. Such efforts would help tea farmers recognize the value and psychological significance of planting insurance, thereby better safeguarding their agricultural interests. This study enhances the understanding of how government support and risk cognition can facilitate the adoption of planting insurance among tea farmers. However, I noticed the modifications and enhancements made by the researchers which are satisfied.

Reviewer #2: (No Response)

7. PLOS authors have the option to publish the peer review history of their article (what does this mean? ). If published, this will include your full peer review and any attached files.

**Do you want your identity to be public for this peer review?** For information about this choice, including consent withdrawal, please see our Privacy Policy .

Reviewer #1: No

Reviewer #2: No

---

## [Editor Report · Acceptance letter]

PONE-D-24-48851R1

PLOS ONE

Dear Dr. Rojniruttikul,

I'm pleased to inform you that your manuscript has been deemed suitable for publication in PLOS ONE. Congratulations! Your manuscript is now being handed over to our production team.

Kind regards,

on behalf of

Professor Xufeng Cui

Academic Editor

PLOS ONE